# Butyrylcarnitine Elevation in Newborn Screening: Reducing False Positives and Distinguishing between Two Rare Diseases through the Evaluation of New Ratios

**DOI:** 10.3390/biomedicines11123247

**Published:** 2023-12-07

**Authors:** MariaAnna Messina, Alessia Arena, Riccardo Iacobacci, Luisa La Spina, Concetta Meli, Federica Raudino, Martino Ruggieri

**Affiliations:** 1Expanded Newborn Screening Laboratory, Newborn Screening and Metabolic Diseases Unit, University-Polyclinic “G. Rodolico-San Marco”, 95123 Catania, Italy; alessia.arena@gmail.com (A.A.); riki713@hotmail.com (R.I.); luisalaspina@hotmail.it (L.L.S.); cmeli@policlinico.unict.it (C.M.); federica.raudino@tiscali.it (F.R.); m.ruggieri@unict.it (M.R.); 2Unit of Clinical Pediatrics, Department of Clinical and Experimental Medicine, University of Catania, 95123 Catania, Italy

**Keywords:** inherited disorders, newborn screening, tandem mass spectrometry, biomarkers

## Abstract

One of the main challenges of newborn screening programs, which screen for inherited metabolic disorders, is cutting down on false positives (FPs) in order to avoid family stresses, additional analyses, and unnecessary costs. False positives are partly caused by an insubstantial number of robust biomarkers in evaluations. Another challenge is how to distinguish between diseases which share the same primary marker and for which secondary biomarkers are just as highly desirable. Focusing on pathologies that involve butyrylcarnitine (C4) elevation, such as short-chain acylCoA dehydrogenase deficiency (SCADD) and isobutyrylCoA dehydrogenase deficiency (IBDD), we investigated the acylcarnitine profile of 121 newborns with a C4 increase to discover secondary markers to achieve two goals: reduce the FP rate and discriminate between the two rare diseases. Analyses were carried out using tandem mass spectrometry with whole blood samples spotted on filter paper. Seven new biomarkers (C4/C0, C4/C5, C4/C5DC\C6OH, C4/C6, C4/C8, C4/C14:1, C4/C16:1) were identified using a non-parametric ANOVA analysis. Then, the corresponding cut-off values were found and applied to the screening program. The seven new ratios were shown to be robust (*p* < 0.001 and *p* < 0.01, 0.0937 < ε^2^ < 0.231) in discriminating between FP and IBDD patients, FP and SCADD patients, or SCADD and IBDD patients. Our results suggest that the new ratios are optimal indicators for identifying true positives, distinguishing between two rare diseases that share the same primary biomarker, improving the predictive positive value (PPV) and reducing the false positive rate (FPR).

## 1. Introduction

Since newborn screening programs for inherited metabolic disorders (IMDs) are catching on worldwide, the prevalence of the diseases is changing, and new pathologies with different degrees of severity are starting to emerge. Oftentimes, alterations in a metabolite can be indicative of the dysfunction of different metabolic pathways, and the magnitude of alterations can be related to the severity of the disease. However, one of the main challenges of newborn screening programs is to cut down on false positives in order to avoid family stresses regarding newborns, additional analyses, and unnecessary costs [1,2,3,4]. The surplus of false positives is partly due to an insubstantial number of robust biomarkers considered during the evaluation of a complete metabolic profile [5,6].

On the other hand, another important goal is the ability to discriminate between rare diseases that involve the alteration of the same primary marker.

Short-chain acyl-CoA dehydrogenase deficiency (SCADD) is a rare autosomal recessive genetic disorder belonging to a group of fatty acid oxidation disorders (FODs). It occurs due to a deficiency of the short-chain acyl-CoA dehydrogenase (SCAD) enzyme. Although newborns with episodes of vomiting, low blood sugar, and fatigue were described, many other asymptomatic newborns have been identified through the expanded newborn screening programs. Thus, SCADD is now viewed as a biochemical phenotype rather than a disease.

IsobutyrylCoA dehydrogenase deficiency (IBDD) is a rare autosomal recessive metabolic disease involving defects in valine catabolism. Patients with IBDD are described to be either asymptomatic or symptomatic with variable clinical features, including seizures, anemia, failure to thrive, or muscular hypotonia. As with SCADD, most patients with IBDD remain healthy for life.

Our newborn screening laboratory screened babies from eastern Sicily for 44 IMDs using tandem mass spectrometry (Appendix A). The alteration in butyrylcarnitine (C4), which is the primary marker for either SCAD or IBDD, is very common in our population, as well as in other populations [7,8,9,10], causing a large number of FPs.

In order to overcome the issue of FPs, a large number of second-tier tests (STTs) were developed. This involved the analysis of metabolites, which were different from those sought as biomarkers in the first test, employing the same sample used for the first test. Therefore, the principal goal of an STT is to drive down the number of FPs, enhancing the sensitivity and specificity of the screening test and avoiding unnecessary parental anxiety. In fact, clinicians recommend recalling a baby for a diagnostic assessment only when the STT gives a positive result; otherwise, the case is closed. On the one hand, the STT supports diagnostic ascertainment without stress for families; on the other hand, it represents an additional assay, which often includes HPLC-UHPLC-MS or NGS analyses [11,12,13,14].

In the specific case of C4′ alteration, the STT involves the search for ethylmalonic acid and isobutyrylglycine for SCADD and IBDD, respectively [15].

A UPLC-MS/MS method has been also developed for the simultaneous quantitation of isobutyrylcarnitine and butyrylcarnitine in order to provide fast differential diagnosis of the two disorders [16,17].

Recently, post-analytical tools have also been developed. However, they require the use of different clinical and anamnestic data, which are not always available and do not always take into account the ratios between metabolites [18,19,20,21].

Between the two pathologies, SCADD certainly gives rise to a greater number of recall tests due to the presence of two common variants, c.511C>T in exon 5 and c.625G>A in exon 6 of the ACADS gene, which result in a biochemical phenotype without any clinical relevance. For this reason, babies, who are either homozygous for a common variant on both ACADS alleles or compound heterozygous for a pathogenic variant on one allele and a c.511C>T or c.625G>A variant on the other allele, are viewed as a biochemical phenotype rather than an affected individual [8,22,23]. Moreover, non-metabolic causes, such as therapeutic hypothermic treatment or the intake of acetylsalicylic acid [24,25], may lead to C4 elevation.

For this purpose, focusing on pathologies that involve C4 alteration, we investigated the metabolic profile of 121 newborns in order to uncover secondary markers for identifying SCAD and IBD deficiencies, reducing false positive rates and discriminating between the two rare diseases.

The 121 babies were recalled to re-evaluate their acylcarnitine profiles. Twenty-six of them were confirmed as SCADD-affected babies, whereas four babies were diagnosed with IBDD through molecular analysis (Table 1). The statistical analysis of their metabolic profiles allowed for finding new ratios, which appeared to be able to discriminate between FPs and TPs since the first screening test. The C4 value alone is not enough guarantee the establishment that the sample is a true or a false positive. For this purpose, once no significant difference was revealed between the metabolites investigated during the first samples run, we explored all possible C4 to acylcarnitines ratios included in our mandatory biomarkers screening panel, and we found seven new ratios as potential secondary biomarkers (Appendix A).

All of them robustly distinguished IBDD patients from FPs. However, only two out of the seven ratios (C4/C5 and C4/C6) were statistically significant to discriminate between SCADD patients from FPs. The ratios exhibited a much stronger discriminatory ability when the SCADD population was divided into three subsets based on genetic results, providing a clear understanding of which SCADD group behaves similarly to the FP group (Table 2 and Table 3).

Five of the new markers were found to have the ability to distinguish SCADD subsets patients from IBDD patients with excellent significance (Table 3).

Furthermore, the same seven ratios proved to be strong biomarkers during the recall test (Table 3).

In order to apply these novel ratios for assessing metabolic profiles of the screened newborns, cut-off values for the new seven ratios were established.

## 2. Materials and Methods

Blood samples were collected from infants within 48 to 72 h after birth for initial screening and within 15 days after birth for recall testing. The samples were spotted onto a Whatman 903TM filter paper (Eastern Business Forms, Greenville, SC, USA) and analyzed using a Neo-base2 TM Non-derivatized MSMS kit (Wallac Oy, Turku, Finland) according to the manufacturer’s instructions, which included the following steps. Using an automatic puncher (Perkin-Elmer Panthera Puncher, Waltham, MA, USA), paper disks (diameter: 3.2 mm) were punched from the DBS into the wells of a plate. A total of 125 μL of a working solution was added to each well of the plate. The plate was stirred at a 750rpm speed in an incubator/shaker (TriNest incubator, Wallac Oy, Turku, Finland) at 30 °C for 30 min. A total of 100 μL of the extracted solution was transferred from each well to another plate.

ESI-MS/MS analyses were carried out on a 210 MD QSightTM spectrometer (Perkin Elmer, Shelton, CT, USA) with a triple quadrupole. Measurements were performed using a QSightTM HC Autosampler MD (Perkin Elmer) and QSightTM binary pump (Perkin Elmer) in the flow injection mode. The injection volume was 10 μL, and the flow rate was 0.02 mL/min. The MS parameters were established as follows: capillary voltage at 5 kV, source temperature at 175 °C, drying gas at 105 L/h, and nebulizer gas at 130 L/h. The analyses were conducted in the multiple reaction monitoring mode (MRM) utilizing stable isotope internal standards to facilitate quantitative analysis. The cut-off criteria were established by utilizing data from a healthy-term newborn population, which represented the 99th percentile. Genetic analyses were conducted on MiniSeq (Illumina, San Diego, CA, USA) using the paired-end 150 bp protocol. Selective enrichment using an amplicon-based strategy (Ampliseq for Illumina, Illumina) preceded sequencing. Sanger sequencing ensured complete coverage of the coding gene.

Confirmatory tests identified 30 true positives and 91 false positives.

We conducted statistical analysis using the Jamovi open statistical software (2.3.28.0) and expressed the measurement data as mean ± SD (standard deviation). Given the low sample numbers, particularly in cases of IBDD where normal distribution assumptions cannot be made, we used non-parametric Kruskal–Wallis’ and Tukey’s post hoc tests to conduct our statistical analysis. Thirty-five C4 ratios were explored with corresponding acylcarnitines (Appendix A). The levels of significance were set at *p* < 0.05, and the χ2 test and ε2 were calculated [26]. Additionally, cut-off values for a neonatal population were evaluated and established in the Cut-Off Analyzer Software^TM^ version 1.11 (2018) (Perkin-Elmer). All concentration values are expressed in μmol/L (μM).

## 3. Results

### 3.1. Newborn Screening Results

Since the mandatory screening program began in eastern Sicily until 31 December 2022, our laboratory screened 105,013 babies, including premature (13,403) and full-term (91,610) babies. Over the same period, 805 babies were recalled and 102 diagnoses (Appendix A) of inherited metabolic diseases were confirmed at the molecular level. Nineteen of these disorders were connected to individuals from the index case family. The largest number of recalls resulted from a C4 alteration, and sure enough, 121 out of 805 recalled babies showed increased levels of this marker, although only 30 were confirmed to be true positives. The average recall rate was 0.80. Our referral center for inherited metabolic diseases closely monitored all diagnosed patients.

### 3.2. SCADD Results

The findings revealed a comparable prevalence of SCADD to that of hyperphenylalaninemia, a commonly occurring condition among rare diseases. In our study population, SCADD diagnoses made up 24% of all diagnoses. The genetic analysis confirmed SCADD in all the affected infants (Table 1). Nine out of the twenty-six genetically confirmed patients could not be deemed clinically relevant due to their homozygosity for the common variant c.625G>A. Seven babies were compound heterozygous for the common variant, and there was one pathogenic mutation. The remaining infants were homozygous for the common variant and heterozygous for a pathogenic mutation. Following these requirements, we categorized our SCADD patients into three different subsets, named SCADD-HH, SCADD-CH, and SCADD-HHC, respectively, to evaluate their contribution towards distinguishing among false positives and the IBDD population (Table 1). From a metabolic profile perspective, this breakdown has been proven to be optimal for gaining a better understanding of which newborns with SCADD are false positive and which are affected, as identified in the initial screening test. When considering the entire SCADD population, the C4 values were unable to differentiate it from the FP population. However, using the two ratios C4/C5 and C4/C6, SCADD newborns could be distinguished from FP newborns, with *p* values of 0.0016 and 0.0187, respectively. Furthermore, the SCADD subsets exhibited different behavioral patterns. In detail, the primary marker remained not significant, and the C4/C5 ratio could not distinguish between FPs and SCADD-HH or SCADD-CH. However, it could discriminate between FPs and SCADD-HHC. The C4/C6 ratio was able to differentiate among more populations, specifically FPs from SCADD-CH with *p* = 0.0144 and FPs from SCADD-HHC with *p* = 0.00026. We examined the same seven new ratios during the recall test. The 121 babies which showed alterations in C4 in their acylcarnitine profiles were recalled, and their samples were analyzed within 10 days of the recall data. The profile trend became more pronounced in the recall test. C4 became significant for distinguishing between FPs and SCADD-HHC as well as between FPs and SCADD-CH. Neither biomarker could differentiate the subset SCADD-HH from FPs, while SCADD-CH was discriminated by the C4/C0 ratio (*p* < 0.01). An interesting finding was discovered when comparing the SCADD-HHC and FP populations. Specifically, six out of the seven ratios were able to differentiate between the two populations in a robust way with *p* < 0.001 except for C4/C5.

No clinical signs were exhibited by any of the newborns who were genetically confirmed at birth or during the follow-up phase. Rather, the SCADD-HHC newborns had a more notable excretion of ethylmalonic acid, as evidenced through the examination of urinary organic acids.

### 3.3. IBCDD Results

A total of 4 babies out of 121 recalled for C4 alteration were confirmed on a molecular level. The incidence of IBDD diagnoses in our population account edfor 4% of all diagnoses with an approximate incidence of 1:26,000. As the number of positive cases was limited, an analysis of the entire IBDD population was necessary. The findings indicated that the primary marker was not able to differentiate between FPs and IBDD. However, all seven biomarkers demonstrated robustness in distinguishing IBDD from FPs. Specifically, while C4/C5 only exhibited a significance of *p* = 0.0136, both the C4/C0 and C4/C16:1 ratios disclosed a significance of *p* < 0.01, and all other markers exhibited a significance of *p* < 0.001 The outcomes demonstrated an improvement in the recall samples, as six ratios achieved statistical significance and displayed *p* < 0.001 except for C4/C5. None of the genetically confirmed IBDD newborns exhibited any clinical signs at birth or during the follow-up phase. Additionally, none of them exhibited urinary excretion of isobutyryl-glycine.

### 3.4. SCADD Versus IBCDD Results

After assessing the ability of the new biomarkers to distinguish between the SCADD and IBDD populations from FPs, we went on to examine their behavior when comparing the SCADD and the IBDD populations.

In the first test, the primary marker could not distinguish between IBDD and any SCADD subset. However, six ratios were significant in discriminating between IBDD and SCADD-HH, with one (C4/C16:1) having a *p*-value of 0.00606 and five (C4/C0, C4/C5DC\C6OH, C4/C6, C4/C8, and C4/C14:1) having *p*-values of less than 0.01. The same five ratios could discriminate between IBDD and SCADD-CH although with *p*-values less than 0.05, except for C4/C8, which still had a *p*-value of less than 0.001. Lastly, the four ratios C4/C5DC\C6OH, C4/C6, C4/C8, and C4/C14 could differentiate between IBDD and SCADD-HHC with with a significance level of *p* < 0.05, except for the C4/C8 ratio, which demonstrated a significance level of *p* < 0.001. In the recall test, the results were particularly interesting. Firstly, C4 became significant for IBDD versus SCADD-HH. Moreover, when comparing the IBDD population to the SCADD-HH subset, the trend of the biomarkers’ significance appeared to be very similar to that observed in the IBDD and FPs comparison. Additionally, some ratios exhibited a higher degree of significance when comparing SCADD-CH and IBDD with the exception of C4/C0 and C4/C5DC\C6OH, which became no longer statistically significant. The comparison between the SCADD-HHC and IBDD populations manifested more differences. In this case, the significance of the C4/C8 ratio decreased, while the C4/C5DC\C6OH and C4/C14:1 ratios were entirely insignificant. The C4/C16:1 ratio became significant with *p* < 0.05, and the significance of the C4/C6 ratio expanded from *p* = 0.0150 to *p* = 0.00446. Except for the C4/C5 ratio, the effect size increased in the recall phase, ranging from ε^2^ = 0.151 to ε^2^ = 0.312.

### 3.5. Cut-Off Settings

After identifying the seven most significant ratios, we analyzed a population of 919 newborns using the Cut-Off Analyzer Software^TM^ to calculate the cut-offs. We selected the following inclusion criteria: sampling was performed between 48–72 h of life, gestational age between 37–41 weeks, and birth weight between 2500–4500 g. We calculated the cut-offs considering the 99th percentile of the population, obtaining the following values expressed with three significant figures: C4/C0 = 0.0349, C4/C5 = 6.62, C4/C5DC\C6OH = 7.34, C4/C6 = 12.7, C4/C8 = 14.3, C4/C14:1 = 5.82, and C4/C16:1 = 2.56. After obtaining the cut-off values, we retrospectively applied them to both the SCADD and IBDD populations (Table 4). Our analysis showed that the four IBDD patients had all their values exceed the new seven ratio cut-off values. Instead, the SCADD patients behaved differently. Specifically, for the C4/C5 ratio, three out of the nine SCADD-HH, five out of the seven SCADD-CH, and eight out of the ten SCADD-HHC patients, respectively had values above the cut-off. For the C4/C6 ratio, values above the cut-off were present in four out of the nine SCADD-HH, five out of the seven SCADD-CH, and all ten SCADD-HHC patients. Regarding all the seven ratios, only one out of the nine SCADD-HH babies showed high values above the cut-offs. Meanwhile, two out of the seven SCADD-CH babies and four out of the ten SCADD-HHC babies shown all the seven values above the respective cut-offs.

### 3.6. Identification of New Patients

The data analysis that was performed after the cut-off setting disclosed two newborns with high values of C4. The analysis of their metabolic profile, referring to the cut-off values of the new biomarkers, showed distinctive characteristics (Table 5). The newborn with the highest value of the primary marker showed ratios of all the values below the corresponding cut-off, whereas the newborn with the lowest C4 value displayed all ratios above the cut-off. The babies were determined to have SCADD-HH and SCADD-HHC, respectively, through molecular analysis. Specifically, c.366_367del (p.Tyr123Profs*24) was identified in the SCADD-HHC newborn.

## 4. Discussion

Our research emphasizes the dominant role of the C4 alteration in the recall rate. Out of the 805 infants recalled, 121 exhibited an elevated value of C4. However, merely 30 infants were confirmed as truly positive, resulting in a low PPV = 25.64. In this scenario, it is crucial to discover new reliable and highly robust biomarkers in order to reduce the number of FPs and distinguish between the two diseases.

The results demonstrate that relying solely on the C4 value cannot give a comprehensive indication of pathology. The C4 values during recall, while surpassing the set cut-off value, were consistently lower than any values found during the first test, which was in agreement with our previous work [27]. Indeed, since mass spectrometry cannot distinguish between the structural isomers, an alteration in butyrylcarnitine or isobutyryl-carnitine in the total acylcarnitine profile could indicate either SCAD deficiency or IBD deficiency.

SCADD falls under the category of fatty-acid β-oxidation disorders, while IBDD falls under the category of valine catabolism disorders (Figure 1).

IBDD is a quite rare disease with limited data in the literature documenting incidences, ranging from 1:45,466 in Italy to approximately 1:62,599 in China or 1:70,000 in the USA [28,29,30,31]. Otherwise, fatty-acid oxidation disorders appear to be more commonly diagnosed [32,33]. The high number of FPs for the C4 alteration can be attributed to various non-metabolic factors, including hypothermic treatment or acetylsalicylic acid administration. However, it is primarily attributed to the presence of the common variants c.511C>T and c.625G>A [22,23].

The use of ratios can lead to overcoming these issues. However, the ratios that are currently used in the main screening panels are not strong enough to significantly reduce the number of FPs. For SCADD, the ratios of C4 to C0 and C2 are widely used along with the ratio of C4 to C3 for IBDD [30]. In 2019, Wang described the C4/C5DC\C6OH ratio as a uniquely better indicator for SCADD [6]. It provides an additional parameter to achieve the goal of reducing FPs. However, there was no evidence that the marker can discriminate against IBDD patients from the first screening test phase. Our study reports, for the first time, a set of robust biomarkers that can reduce FPs and distinguish amongst SCADD and IBDD patients from the first screening test. Of the two pathologies, SCADD was the one that was the least different from the population of FPs, with only the two ratios C4/C5 and C4/C6 being found to be statistically significant. It is noteworthy that it was only when the SCADD population was divided into three subsets that we could evaluate the contribution of each group to the significance of the obtained data. Therefore, we discovered that the SCADD-HH subset cannot be differentiated from the FP population, indicating that variant c.625G>A does not contribute to a significant alteration to the metabolic profile. On the other hand, the SCADD-HHC subset was the group that differed more significantly from the FP population. The subset SCADD-CH showed an intermediate condition. Our results indicate that a significant change in metabolic profile occurred solely when the common variant was linked with at least one pathogenic variant. In fact, the number of newborns with all ratio values exceeding the cut-offs increased from the SCADD-HH subset, with one out of the nine babies (11.1%), to the SCADD-CH subset, with two out of the seven babies (28.6%), up to the SCADD-HHC subset, with four out of the ten babies (40.0%). The four SCADD-HHC babies who showed alterations in all seven ratios were characterized by the mutation c.1147C>T, which clearly confers a significant pathogenicity when coupled to the benign variant c.625G>A [34]. On the contrary, if the increase in C4 during the first test is coupled with an alteration in all seven ratios with very high values, it indicates a strongly altered metabolic profile and that we may be facing an IBDD profile. The IBDD population behaved very differently from the FP as well as from SCADD-HH subset, confirming that this one looks like a false positive.

Moreover, it is worthy of note that all the IBDD patients showed values of the seven new ratios that were abundantly higher than the set cut-offs. The data confirm that SCADD-HHC group diverged better from the FP population. In fact, eighty percent and one hundred percent of the babies in this group had values higher than the cut-off for the C4/C5 and C4/C6 ratios, respectively. Conversely, the other subset had much lower percentages.

These two ratios were the only two markers which disclosed a large size effect (ε^2^ = 0.168 and ε^2^ = 0.231, respectively) in the first test. Despite its mild significance among the groups, the C4/C5 ratio resulted in a large effect size and was distinctive for SCADD. All the other five new ratios showed a medium effect size, whereas C4 revealed a small effect size, ε^2^ = 0.008 (Table 3). The data obtained from the newborns investigated after the cut-off setting were consistent with our argument. In fact, the baby with low ratio values was easily identified as SCADD-HH or false positive, while a baby with high values of all the ratios could be considered as SCADD-HHC or IBDD. However, when analyzing the absolute values of the ratios, they did not seem as to be high as those of the IBDD newborns. Therefore, our data interpretation of the metabolic profile of the first screening test, without genetic results, was leaning toward the SCADD-HHC group. To confirm our hypothesis and assure the validity of the new markers, we recalled the newborns and performed genetic analyses. As assumed, the two babies were confirmed as SCADD-HH (homozygous for the common variant c.625G>A) and SCADD-HHC (homozygous for the common variant c.625G>A, and heterozygous for the pathogenetic mutation c.366_367del), respectively. To the best of our knowledge, this study is the first to outline seven ratios that can reduce the false positive rate and distinguish between two rare diseases that share the same primary marker from the first screening test. This approach can limit costs by reducing the number of second-tier tests and recalls required. Additionally, the early identification of SCADD or IBDD would enable clinicians to adopt a tailored approach towards the family based on the type of disorder. Unfortunately, our study did not include homozygotes or compound heterozygotes for two pathogenetic variants in SCADD patients, nor did it include a significant number of IBDD patients. Therefore, we lack knowledge regarding how they compare to the new seven markers we discovered. For this purpose, a subsequent study, possibly multicentric, would be desirable.

## Figures and Tables

**Figure 1 biomedicines-11-03247-f001:**
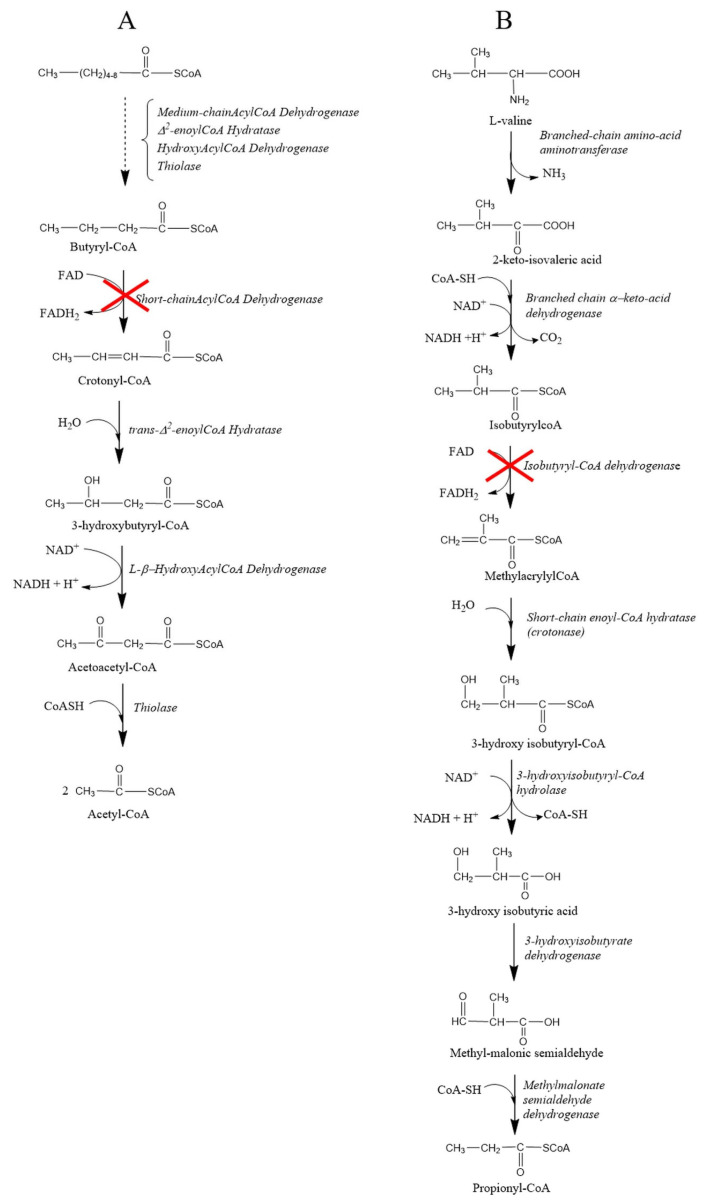
Metabolic pathways: (**A**) medium-/short-chain fatty-acid β-oxidation; (**B**) catabolic pathway of valine. The red crosses indicate the metabolic block that leads to SCADD and IBDD, respectively.

**Table 1 biomedicines-11-03247-t001:** Patients’ biochemical and genetic features.

Patient	Disease	C4 (µM) First Test (Cut-Off 0.62 µM)	C4 (µM) Recall (Cut-Off 0.35 µM)	Gene	Mutation
1	SCADD-HH	1.22	0.17	*ACADS*	c.625G>A homozygous
2	SCADD-HH	1.65	0.30	*ACADS*	c.625G>A homozygous
3	SCADD-HH	0.81	0.20	*ACADS*	c.625G>A homozygous
4	SCADD-HH	0.90	0.27	*ACADS*	c.625G>A homozygous
5	SCADD-HH	0.89	0.17	*ACADS*	c.625G>A homozygous
6	SCADD-HH	0.94	0.37	*ACADS*	c.625G>A homozygous
7	SCADD-HH	0.73	0.36	*ACADS*	c.625G>A homozygous
8	SCADD-HH	0.90	0.27	*ACADS*	c.625G>A homozygous
9	SCADD-HH	1.06	0.39	*ACADS*	c.625G>A homozygous
1	SCADD-CH	1.08	0.52	*ACADS*	c.625G>A heterozygous c.320G>A heterozygous
2	SCADD-CH	1.10	0.56	*ACADS*	c.625G>A heterozygous c.889C>T heterozygous
3	SCADD-CH	0.97	0.42	*ACADS*	c.625G>A heterozygous c.527dup heterozygous
4	SCADD-CH	0.36	0.36	*ACADS*	c.625G>A heterozygous c.310_312del heterozygous
5	SCADD-CH	1.10	0.42	*ACADS*	c.625G>A heterozygous c.988C > A heterozygous
6	SCADD-CH	1.04	0.60	*ACADS*	c.625G>A heterozygous c.814C>T heterozygous
7	SCADD-CH	0.85	0.42	*ACADS*	c.625G>A heterozygous c.268G>A heterozygous
1	SCADD-HHC	0.83	0.38	*ACADS*	c.625G>A homozygous c.1147C>T heterozygous
2	SCADD-HHC	0.99	0.75	*ACADS*	c.625G>A homozygous c.1147C>T heterozygous
3	SCADD-HHC	0.95	0.40	*ACADS*	c.625G>A homozygous c.1147C>T heterozygous
4	SCADD-HHC	1.03	0.44	*ACADS*	c.625G>A homozygous c.136C>T heterozygous
5	SCADD-HHC	0.94	0.31	*ACADS*	c.625G>A homozygous c.366_367delC>T heterozygous
6	SCADD-HHC	0.95	0.51	*ACADS*	c.625G>A homozygous c.1147C>T heterozygous
7	SCADD-HHC	0.82	0.54	*ACADS*	c.625G>A homozygous c.1147C>T heterozygous
8	SCADD-HHC	0.82	0.49	*ACADS*	c.625G>A homozygous c.253C>T heterozygous
9	SCADD-HHC	1.13	0.57	*ACADS*	c.625G>A homozygous c.268G>A heterozygous
10	SCADD-HHC	1.27	0.54	*ACADS*	c.625G>A homozygous c.988C>T heterozygous
1	IBDD	1.42	0.71	*ACAD8*	c.958G>A heterozygous c.1129G>A heterozygous
2	IBDD	1.90	0.62	*ACAD8*	c.512C > G homozygous
3	IBDD	0.94	0.59	*ACAD8*	c.512C > G homozygous
4	IBDD	0.73	0.46	*ACAD8*	c.259G>A heterozygous c.512C > G heterozygous

C4 = butyryl/isobutyrylcarnitine; SCADD-HH = short-chain acylCoA dehydrogenase deficiency—patient homozygous for the common variant c.625G>A; SCADD-HC = short-chain acylCoA dehydrogenase deficiency—patient compound heterozygous for the common variant c.625G>A and another mutation; SCADD-HHC = short-chain acylCoA dehydrogenase deficiency—patient homozygous for the common variant c.625G>A and heterozygous for another mutation; IBDD = isobutyrylCoA dehydrogenase deficiency.

**Table 2 biomedicines-11-03247-t002:** Statistical parameters for SCADD, IBDD, and FPs in the first test.

Marker	Mean (μM)			Standard Deviation (μM)			*p*-Value			χ^2^	ε^2^
	SCADD (*n* = 26)	IBDD (*n* = 4)	FPs (*n* = 91)	SCADD (*n* = 26)	IBDD (*n* = 4)	FPs (*n* = 91)	SCADD-FP	IBDD-FPs	SCADD-IBDD		
C4	0.974	1.25	0.979	0.224	0.522	0.177	0.9933	0.0299	0.0368	0.738	0.006
C4/C0	0.0505	0.0796	0.0461	0.0214	0.0220	0.0151	0.4663	0.00048	0.0048	7.98	0.067
C4/C5	7.55	9.82	6.14	2.76	2.36	2.10	0.0016	0.0052	0.1524	15.5	0.129
C4/C5DC\C6OH	9.56	14.3	8.50	2.00	4.26	2.25	0.0957	5.32 × 10^−6^	0.00044	10.2	0.085
C4/C6	16.1	27.6	13.4	4.13	4.41	4.24	0.0187	1.14 × 10^–13^	1.33 × 10^−8^	25.5	0.213
C4/C8	16.9	33.3	15.5	5.97	11.5	5.59	0.5308	1.00 × 10^−7^	2.78 × 10^−6^	10.4	0.086
C4/C14:1	8.22	14.9	7.54	3.07	3.67	3.04	0.5766	2.26 × 10^−5^	0.00028	10.4	0.087
C4/C16:1	4.14	6.67	3.81	1.41	1.16	1.53	0.5804	0.00086	0.0061	11.2	0.093

SCADD = short-chain AcylCoA dehydrogenase deficiency; IBDD = isobutyryl dehydrogenase deficiency; FPs = false positives; *n* = number of diagnosed babies; χ^2^ = chi-squared test; ε^2^ = effect size; C4 = isobutyrylcarnitine; C0 = free carnitine; C5 = isovalerylcarnitine; C5DC\C6OH = glutarylcarnitine\3-hydroxy-hexanoylcarnitine; C6 = hexanoylcarnitine; C8 = octanoylcarnitine; C14:1 = tetradecenoylcarnitine; C16:1 = hexadecenoylcarnitine; media and standard deviation are expressed with three significant figures.

**Table 3 biomedicines-11-03247-t003:** Statistical parameters for SCADD subsets, IBDD, and FPs in the first test.

Marker	Mean (μM)					Standard Deviation (μM)					*p*-Value							χ^2^	ε^2^
I Test	SCADD-HH (*n* = 9)	SCADD-CH (*n* = 7)	SCADD-HHC (*n* = 10)	IBDD (*n* = 4)	FPs (*n* = 91)	SCADD-HH (*n* = 9)	SCADD-CH (*n* = 7)	SCADD-HHC (*n* = 10)	IBDD (*n* = 4)	FPs (*n* = 91)	SCADD-HH-FPs	SCADD-CH-FPs	SCADD-HHC-FPs	IBDD-FP	SCADD-HH-IBDD	SCADD-CH-IBDD	SCADD-HHC-IBDD		
C4	1.01	0.929	0.973	1.25	0.979	0.278	0.266	0.144	0.522	0.177	0.9918	0.9699	0.9999	0.0844	0.3131	0.1018	0.1644	0.926	0.008
C4/C0	0.0389	0.0502	0.0611	0.0796	0.0461	0.00574	0.0242	0.0244	0.0220	0.0151	0.7238	0.9674	0.0522	0.00105	0.00066	0.0399	0.3221	15.0	0.125
C4/C5	6.19	7.93	8.51	9.82	6.14	1.75	4.22	1.90	2.36	2.10	1.000	0.2486	0.0154	0.0136	0.0586	0.6602	0.8575	20.1	0.168
C4/C5DC\C6OH	8.47	10.1	10.2	14.3	8.50	1.44	2.54	1.78	4.26	2.25	1.000	0.3926	0.1790	1.60 × 10^−5^	0.00032	0.0263	0.0194	13.0	0.109
C4/C6	13.5	18.9	19.7	28.0	13.4	3.20	6.50	4.25	6.08	4.24	1.000	0.0144	0.00026	1.61 × 10^−8^	2.11 × 10^−6^	0.0100	0.0150	27.7	0.231
C4/C8	14.8	17.8	18.2	33.3	15.5	3.31	6.84	7.17	11.48	5.59	0.9972	0.8618	0.6576	3.56 × 10^−7^	8.31 × 10^−6^	0.0005	0.0003	11.2	0.0937
C4/C14:1	7.24	8.85	8.66	14.86	7.54	3.01	3.98	2.46	3.67	3.04	0.9987	0.8111	0.8068	7.84 × 10^−5^	0.00064	0.0187	0.00769	11.8	0.0986
C4/C16:1	3.53	4.70	4.30	6.67	3.81	0.914	2.11	1.07	1.16	1.53	0.9834	0.5541	0.8585	0.00264	0.00606	0.2280	0.0649	13.5	0.112
**Recall**																			
C4	0.278	0.471	0.493	0.595	0.300	0.0850	0.0886	0.122	0.103	0.0983	0.9679	0.00022	4.76 × 10^−7^	5.20 × 10^−7^	4.96 × 10^−6^	0.2782	0.4146	37.5	0.312
C4/C0	0.00950	0.0185	0.0221	0.0247	0.0118	0.00246	0.00800	0.0127	0.00601	0.00408	0.7553	0.01936	1.10 × 10^−6^	0.00011	0.00010	0.3785	0.9347	28.8	0.240
C4/C5	1.37	2.59	2.84	3.69	1.66	0.435	2.10	1.41	2.56	0.812	0.9285	0.1627	0.00796	0.00198	0.00284	0.4414	0.6403	16.5	0.137
C4/C5DC\C6OH	6.22	8.05	9.24	11.8	5.69	1.90	2.89	3.78	4.01	1.95	0.9629	0.06651	6.82 × 10^−5^	6.63 × 10^−6^	0.00078	0.07313	0.3286	22.9	0.191
C4/C6	7.14	10.4	14.8	21.5	7.90	1.78	2.55	5.01	10.5	2.39	0.9576	0.2715	9.32 × 10^−9^	8.21 × 10^−13^	7.34 × 10^−11^	1.05 × 10^−6^	0.00446	35.0	0.292
C4/C8	6.83	9.99	13.7	19.3	7.64	2.77	2.88	3.83	6.44	2.62	0.9309	0.2458	4.11 × 10^−8^	2.65 × 10^−10^	1.00 × 10^−9^	1.38 × 10^−5^	0.01417	30.3	0.250
C4/C14:1	6.45	10.6	13.8	18.6	8.23	1.76	3.50	3.98	4.30	3.35	0.5513	0.3570	1.91 × 10^−5^	1.74 × 10^−7^	1.95 × 10^−7^	0.00222	0.1201	30.0	0.250
C4/C16:1	5.50	9.47	10.9	17.8	7.89	1.88	5.14	4.25	4.94	4.23	0.4802	0.8703	0.2126	0.00010	3.37 × 10^−5^	0.01676	0.0478	18.1	0.151

SCADD = short-chain AcylCoA dehydrogenase deficiency; IBDD = isobutyryl dehydrogenase deficiency; FPs = false positives; *n* = number of diagnosed babies; χ^2^ = chi-squared test; ε^2^ = effect size; C4 = isobutyrylcarnitine; C0 = free carnitine; C5 = isovalerylcarnitine; C5DC\C6OH = glutarylcarnitine\3-hydroxy-hexanoylcarnitine; C6 = hexanoylcarnitine; C8 = octanoylcarnitine; C14:1 = tetradecenoylcarnitine; C16:1 = hexadecenoylcarnitine; media and standard deviation are expressed with three significant figures.

**Table 4 biomedicines-11-03247-t004:** Values of new biomarkers in the diagnosed patients.

Patient	Disease		Ratio—First Test					
		C4/C0 (Cut-Off 0.0349)	C4/C5 (Cut-Off 6.62)	C4/C5DC\C6OH (Cut-Off 7.34)	C4/C6 (Cut-Off 12.9)	C4/C8 (Cut-Off 14.3)	C4/C14:1 (Cut-Off 5.82)	C4/C16:1 (Cut-Off 2.56)
1	SCADD-HH	0.0462	3.48	9.38	12.2	8.73	3.05	3.21
2	SCADD-HH	0.0362	9.16	8.25	20.6	12.6	13.7	3.05
3	SCADD-HH	0.0336	6.23	8.10	11.5	16.2	7.36	2.38
4	SCADD-HH	0.0315	6.92	6.42	10.0	15.0	7.50	3.91
5	SCADD-HH	0.0441	8.09	8.90	14.8	17.8	8.09	3.29
6	SCADD-HH	0.0470	5.52	8.54	15.6	18.8	5.22	4.08
7	SCADD-HH	0.0384	6.08	10.4	12.1	14.6	8.11	4.05
8	SCADD-HH	0.0396	4.28	10.0	11.2	18.0	7.50	5.29
9	SCADD-HH	0.0337	5.88	6.23	13.2	11.7	4.60	2.46
1	SCADD-CH	0.0279	8.30	7.20	21.6	15.4	9.81	2.76
2	SCADD-CH	0.0724	15.7	12.2	22.0	18.3	10.0	4.40
3	SCADD-CH	0.0516	7.46	8.08	12.1	12.1	3.73	3.12
4	SCADD-CH	0.0120	1.89	7.20	9.00	12.0	5.14	9.00
5	SCADD-CH	0.0786	8.46	12.2	18.3	12.2	6.11	3.92
6	SCADD-CH	0.0654	8.66	13.0	20.8	26.0	13.0	4.00
7	SCADD-CH	0.0436	5.00	10.6	28.3	28.3	14.1	5.66
1	SCADD-HHC	0.0546	9.22	11.8	27.6	27.6	11.8	5.18
2	SCADD-HHC	0.0723	9.00	11.0	19.8	16.5	11.0	5.50
3	SCADD-HHC	0.0432	9.50	9.50	19.0	11.8	7.30	3.39
4	SCADD-HHC	0.0200	9.36	11.4	20.6	12.8	6.86	3.67
5	SCADD-HHC	0.0573	7.83	9.40	18.8	10.4	5.22	2.93
6	SCADD-HHC	0.0931	7.91	7.91	15.8	19.0	5.93	3.51
7	SCADD-HHC	0.0453	7.45	11.7	20.5	20.5	11.7	3.03
8	SCADD-HHC	0.0582	5.85	11.7	13.6	9.11	7.45	5.12
9	SCADD-HHC	0.1046	12.5	6.64	16.1	28.2	8.69	5.13
10	SCADD-HHC	0.0626	6.35	10.5	25.4	25.4	10.5	5.52
1	IBDD	0.1044	12.9	17.7	28.4	47.3	20.2	7.88
2	IBDD	0.0888	10.0	15.8	23.7	38.0	13.5	6.33
3	IBDD	0.0718	7.23	15.6	23.5	23.5	13.4	7.23
4	IBDD	0.0533	9.12	8.11	36.5	24.3	12.1	5.21

C4 = isobutyrylcarnitine; C0 = free carnitine; C5 = isovalerylcarnitine; C5DC\C6OH = glutarylcarnitine\3-hydroxy-hexanoylcarnitine; C6 = hexanoylcarnitine; C8 = octanoylcarnitine; C14:1 = tetradecenoylcarnitine; C16:1 = hexadecenoylcarnitine; SCADD-HH = short-chain acylCoA dehydrogenase deficiency—patient homozygous for the common variant c.625G>A; FPs = false positives; SCADD-CH = short-chain acylCoA dehydrogenase deficiency—patient compound heterozygous for the common variant c.625G>A and another mutation; SCADD-HHC = short-chain acylCoA dehydrogenase deficiency—patient homozygous for the common variant c.625G>A and heterozygous for another mutation; IBDD = isobutyrylCoA dehydrogenase deficiency; in brackets, the new cut-off values found. Values are expressed with three significant figures.

**Table 5 biomedicines-11-03247-t005:** Biochemical and genetic features of diagnosed newborns performed after cut-off setting.

Patient	Disease Mutation	Primary Marker	Ratio—First Test								
			C4 First Test (Cut-Off 0.62 μM)	C4 Recall (Cut-Off 0.35 μM)	C4/C0 (Cut-Off 0.0349)	C4/C5 (Cut-Off 6.62)	C4/C5DC\C6OH (Cut-Off 7.34)	C4/C6 (Cut-Off 12.9)	C4/C8 (Cut-Off 14.3)	C4/C14:1 (Cut-Off 5.82)	C4/C16:1 (Cut-Off 2.56)
1	SCADD-HH	c.625G>A (homozygous)	0.943	0.361	0.0281	6.27	6.28	5.86	5.88	2.47	1.68
2	SCADD-HHC	c.625G>A (homozygous) c.366_367del (heterozygous)	0.862	0.522	0.0422	7.17	10.7	21.5	17.2	8.60	4.77

C4 = isobutyrylcarnitine; C0 = free carnitine; C5 = isovalerylcarnitine; C5DC\C6OH = glutarylcarnitine\3-hydroxy-hexanoylcarnitine; C6 = hexanoylcarnitine; C8 = octanoylcarnitine; C14:1 = tetradecenoylcarnitine; C16:1 = hexadecenoylcarnitine; SCADD-HH = short-chain acylCoA dehydrogenase deficiency—patient homozygous for the common variant c.625G>A; FPs = false positives; SCADD-CH = short-chain acylCoA dehydrogenase deficiency—patient compound heterozygous for the common variant c.625G>A and another mutation; SCADD-HHC = short-chain acylCoA dehydrogenase deficiency—patient homozygous for the common variant c.625G>A and heterozygous for another mutation. Values are expresses with three significant figures.

## Data Availability

Data will be made available on request.

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
