# Peer review of "Butyrylcarnitine Elevation in Newborn Screening: Reducing False Positives and Distinguishing between Two Rare Diseases through the Evaluation of New Ratios"

_biomedicines, 2023, doi:10.3390/biomedicines11123247_

Round 1

Reviewer 1 Report

Comments and Suggestions for Authors

Rare metabolic disorders are sometimes difficult to detect. Newborn screening programs have improved the detection of new cases, unfortunately false positives are also described. Better biomarkers are mandatory, not only for diminish false positives but also to discriminate on diseases phenotypes. This article from a newborn screening center presents a deeply analyze of 121 newborns regarding SCADD and IBDD disorders, which resulted in the identification of new biomarkers an important asset for clinicians in metabolic centers. Hopefully more studies like this will be performed to corroborate this findings, and should be spanned to other metabolic disorders. 

Comments on the Quality of English Language

The manuscript would benefit of an overall English language revision.

Author Response

We would like to thank the reviewer for his thoughtful review of the manuscript. We really appreciated her/his comment. English language was revised in the final revision.

Reviewer 2 Report

Comments and Suggestions for Authors

In their manuscript "Butyrylcarnitine elevation in newborn screening: reducing false positive and distinguishing between two rare diseases through evaluation of new ratios." the authors describe in detail, the introduction of new ratios to distinguish true positve (tp) cases of SCADD and IBDD, from false positives (fp). In addition the autors also mention possible second tier tests (STT).

However, there several major drawbacks of the manuscript:

1) The authors do not provide any reason, why they did not use these STT in the first place to reduce fp. In addition STT cannot only determine ethylmalonic acid and isobutyrylglycine.

2) With the same STT, that differentiates isovalerylcarnitine form methylbutyrylcarnitine and pivaloylcarnitine, also butyrylcarnitine can be differentiated from isobutyrylcarnitine.

3) The authors should describe in detail the clinical penetrance, and therfore also the clinical relevance of newborn screening for SCADD, and IBDD. It is generally accepted that these enzyme deficiencies are rather non-diseases.

4) The authors do not comment on the fact, that most of their "tp" cases have normal C4 concentrations in the 2nd sample.

5) The authors only provide genetic data, which are used to classify babies with elevated C4 concentrations as tp or fp. However, non of the cases is homozygous or compound hetreozygous for a "pathogenic" variant. They are only compound heterozygous in connection  with the common variant.

6) To prove that a tp is really a tp, enzymatic analysis of the SCAD-gene is necessary.

7) No information is provided on the clinical signs of the tp cases. Hypoglycaemia, organic acids in urine, etc.

Author Response

Response 1: We would like to thank the reviewer for his thoughtful review of the manuscript. However, the authors are keen to point out that our work does not conflict with the execution of second tier tests. Any laboratory wishing to perform a second tier confirmatory test can obviously do so. Our work aims to offer an additional tool, robust and reliable, to be able to understand what type of deficiency you are dealing with, right from the first test. Our work therefore strengthens diagnostic tools, not intended to replace them.

Response 2: According to the suggestion, we have included a reference on the execution of other second tiers that help the diagnostic path for IBDD and SCADD.

Response 3: Thank you for the comment. However, we would like to remind you that the authors declare the clinical non-relevance for SCADD characterized by the benign variant c.625G>A. For other conditions, especially for IBDD, although there are many asymptomatic cases, cases with clinical signs are described, as reported in the references 30 and 31 we have cited.

Furthermore, we want to point out that our work does not want to analyze the clinical aspect of the conditions. It only wants to focus on the biochemical-analytical aspect, in order to help technical staff and clinicians to better understand what type of alteration they are faced with, starting from a single test. In fact, from an analytical point of view, an alteration of biomarkers cannot be overlooked.

Response 4: Thank you for the comment. We had discussed the decrease in the value of C4 in a previous work. In accordance with what was suggested we inserted a sentence in the text and added the corresponding reference.

Response 5: Thank you for the comment. We are aware that our cases do not include subjects with two pathogenic mutations and we declare this in the text. Our work serves precisely to understand how much impact the subjects with a pathogenetic mutation coupled with the common variant. That is, it serves to better understand how this genetic condition can alter the biochemical profile and consequently help laboratory personnel understand which condition they are dealing with from the first screening test.

Response 6: As widely accepted by all screening laboratories and by the Italian scientific society on inherited metabolic diseases and neonatal screening (SIMMESN, https://www.simmesn.it/it/documenti/rapporti-tecnici-screening-neonatale.html ), when a suspicion of positivity at screening occurs, the case is confirmed with the analysis genetics. Enzymatic analysis of the enzyme encoded by the ACADS gene is performed if there are inconsistencies between clinical signs and genetic analysis. In our case, no newborn was symptomatic and none had two pathogenic mutations. We will consider enzymatic analyses regardless in the future.

Response 7: As previously mentioned, although our work is not focused on clinical features, we commented in the text on the main clinical characteristics of positive newborns.

Reviewer 3 Report

Comments and Suggestions for Authors

Dear Editor,

I would like to express my deep thanks to you for allowing me to review this valuable manuscript, Butyrylcarnitine elevation in newborn screening: reducing false positive and distinguishing between two rare diseases through evaluation of new ratios". This article examined the metabolic profile of 121 newborns, concentrating on pathologies involving C4 modification, in an effort to develop secondary indicators for SCAD and IBD deficiencies, lower the false positive rate, and improve the ability to distinguish between the two rare disorders.

.

·         In the introduction section, the full term of IBDD, SCAD, C4, and FP should be written

·         SCAD and IBD need to have more background information presented in the introduction

·         The authors created a beautiful figure, but the figure in the manuscript is not clear enough. The original or high-definition figures should be provided

·         Scientific articles with several technical terms or acronyms should include an abbreviation list. Abbreviations can improve article reading and comprehension

·         Reference style according to Biomedicine journal format should be placed in square brackets [ ]

Author Response

We would like to thank the reviewer for his thoughtful review of the manuscript. We really appreciated her/his comment.

Response 1: According to the suggestion, we wrote the names in full.

Response 2: According to the suggestion, we have added sentences about SCAD and IBD background information.

Response 3: We have attached a high definition image in the text and we have loaded it separately during the submission process.

Response 4: According to the suggestion, we have included an abbreviation list.

Response 5: We apologize for the formatting error. We have made the corrections.
